# Determinants of Contraceptive Use among Unmarried Young Women in Kakamega County, Kenya

Elizabeth Arlotti-Parish [1,*], Carolyne Ajema [2], Lilian Mutea [3] and Susan Ontiri [4]

1   Jhpiego, Baltimore, MD 21231, USA
2   Jhpiego, Nairobi P.O. Box 66119-00800, Kenya; carol.ajema@jhpiego.org
3   United States Agency for International Development, Nairobi P.O. Box 629-00621, Kenya; lmutea@usaid.gov
4   International Center for Reproductive Health Kenya, Mombasa P.O. Box 91109-80103, Kenya; sontiri@icrhk.org
*   Correspondence: elizabeth.arlotti-parish@jhpiego.org

**Abstract:** Adolescent pregnancies adversely impact mental and reproductive health as well as educational and socio-economic outcomes. In Kakamega County, Kenya, 20% of adolescents begin childbearing by age 19. To inform interventions to reduce adolescent pregnancy, Jhpiego used the Barrier Analysis methodology, which is based on the Doer/Non-Doer study model, in which participants are categorized according to whether they are "Doers" or "Non-Doers" of the study behavior. This study examines the determinants of the behavior, "young unmarried women currently use modern contraceptive methods". Participants included young women aged 15–19 who were sexually active, unmarried, and were using ("Doers") or not using ("Non-Doers") modern contraception. The findings reveal that the majority of Doers (88%) and Non-Doers (80%) understand the pregnancy risk associated with non-use, and there is no statistically significant difference between Doers' and Non-Doers' understanding of contraceptive benefits. Knowledge of side effects and misconceptions, such as the belief that contraception causes infertility, does not deter Doers from using contraception. Seventy percent of Doers note that contraception is accessible/available, while 39% of Non-Doers state the opposite. Doers are almost three times more likely than Non-Doers to say that most people approve of their contraceptive use, while Non-Doers are twice as likely as Doers to say that most people would not approve. Doers are four times more likely to indicate approval from their mothers and boyfriends. Non-Doers are five times more likely than Doers to have specific professional goals for the future. These findings illustrate the importance of moving away from fear-based messaging and instead highlighting social acceptability and contraception's role in achieving future goals.

**Keywords:** gender; contraception; family planning; Kakamega; Kenya; adolescent; barrier analysis; social norms; Doer; Non-Doer

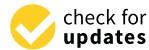



## 1. Introduction

Adolescents aged 10 to 19 constitute 16% of the world's population [1]. Global estimates indicate that 21 million pregnancies occur annually among 15–19-year-olds in low- and middle-income countries [2]. The majority of these pregnancies are unintended [3]. Research shows that pregnancies adversely impact adolescents' mental and reproductive health as well as their educational and socio-economic outcomes [4,5]. This can stem from complications resulting from an unsafe abortion, rapid repeat pregnancies, and complications during and after delivery [6–8]. In addition, the earning power, decision-making, and agency of teen mothers is limited, and this contributes to the inequitable outcomes between boys and girls [9–12].

According to the 2019 Kenya Population and Housing Census, young people below the age of 15 constitute 39% of the total population (47.6 million) [13]. In Kenya, adolescents experience several challenges that impede their health outcomes and well-being, thus increasing their vulnerability [14] to early sexual debut, early and unintended pregnancy,

sexually transmitted infections, unsafe abortions, gender-based violence, and child marriages. According to the United Nations, 40% of girls in sub-Saharan Africa marry before 18 [15]. Teenage pregnancy and harmful sociocultural practices constitute some of the determinants of early marriage [16–18]. The increased occurrences of pregnancies among 15–19-year-olds demonstrate lower contraceptive use. The Kenya Demographic and Health Survey shows that 15% of women aged 15–19 have been pregnant and have a 22% unmet need for family planning, compared with 14% for all women of reproductive age [19].

Kenya's National Adolescent Sexual and Reproductive Health policy outlines adolescents' sexual and reproductive rights and a framework to govern service provision [6]. However, contraceptive use among this population remains low at 40% [19]. Investments in reproductive and maternal health have not translated into equitable access to or use of services. Factors contributing to the high rates of pregnancy among adolescents, both globally and in Kenya, include inadequate adolescent-responsive sexual and reproductive health (SRH) policies, limited knowledge of available services, providers' attitudes towards adolescents seeking SRH services (including contraceptives), the inadequacy of age-appropriate SRH information, low decision-making power among adolescents, and harmful social norms [5,20]. A qualitative study exploring the barriers to modern contraceptive use among unmarried women in Kenya notes fear of side effects and misconceptions about adverse reaction as major barriers to use [21]. However, this study was limited to urban and peri-urban populations and included diverse populations, including adolescents and young adults up to age 24.

Other underlying barriers to the uptake of modern contraceptives by adolescents have yet to be explored. Our study assessed the barriers and enablers related to the uptake of modern contraception among unmarried adolescents aged 15–19 years in Kakamega County, a rural county that was recommended for the study by county leaders who were concerned about its high rate of adolescent pregnancy. At the time of the study, Kakamega had the lowest contraceptive uptake among 15–24-year-olds of any county in Kenya at 1%, and 20% of women in the county had begun childbearing by the age of 19 [22]. Understanding the behaviors of adolescents regarding the use or non-use of contraceptives can guide service providers and policymakers in determining the appropriate approaches towards reducing unplanned pregnancies and helping young people to achieve their reproductive intentions.

This study was conducted as part of a broader U.S. Agency for International Development-funded Afya Halisi ("real health" in Swahili) Project, whose goal was to improve the utilization of quality reproductive, maternal, newborn, child, and adolescent health, nutrition, and water hygiene and sanitation services in four counties in Kenya (Kitui, Kakamega, Migori, and Kisumu).

## 2. Materials and Methods

### 2.1. Study Procedures and Measures

Our study was conducted as part of a larger gender analysis aimed at understanding how gender norms and cultural practices impact women's and men's experiences, perceptions, and utilization of services in four counties in Kenya. The relevant objectives of the larger gender analysis explored through this study were to identify gender norms and cultural practices that are barriers to family planning service uptake, and to assess gender-based disparities across age groups in the use of modern contraception services.

This study used the Barrier Analysis methodology, a formative research tool developed as part of the Designing for Behavior Change Framework and based on prior Doer/Non-Doer methodologies. The Barrier Analysis methodology draws from existing behavioral models, including the theory of reasoned action, health belief, and perceived behavioral control models [23]. The Barrier Analysis methodology has been primarily used to study health behaviors, though has been expanded to support the exploration of determinants of agricultural behaviors. The methodology is designed to identify determinants of a specific behavior among a relatively homogeneous group using a standardized questionnaire

format in which each question represents one major determinant category and invites the inclusion of specific content related to the key behavior being examined (Table 1).

**Table 1.** Key determinants of health behaviors.

| Key Determinant | Definition |
| --- | --- |
| Perceived Self-Efficacy/Skills | A person's belief in their ability to do the behavior given the time, knowledge, skills, and resources available to them. |
| Perceived Social Norms | A person's belief in the social acceptability of doing the behavior. |
| Perceived Positive or Negative Consequences | A person's belief in the advantages or disadvantages of doing the behavior, or the positive or negative outcomes that will occur as a result of doing the behavior. |
| (Perceived) Access | A person's belief in their access to products or services needed to do the behavior. |
| Perceived Barriers | Things that make it more difficult to do the behavior. |
| Perceived Enablers | Things that make it easier to do the behavior. |
| Cues for Action | The presence of cues to help a person remember to do a behavior. |
| Perceived Susceptibility | A person's perceived vulnerability to something bad that could happen to them if they do not do the behavior. |
| Perceived Severity | The perceived seriousness of a negative outcome that could occur if a person does not do the behavior. |
| Perception of Divine Will | A person's belief that God (or the gods) wants them to have a problem or overcome a problem associated with doing or not doing a behavior. |
| Policy | Laws and regulations that affect a behavior and access to products and services needed to do the behavior. |
| Culture | The history, customs, lifestyles, values, and practices within a self-defined group that influence behavior. Culture often influences perceived social norms. |
| Universal Motivators | The set of motivators that influence most people, regardless of context. These include love, success, wealth, happiness, and others, and are often used by corporate advertisers to reach large, diverse populations. |

It is important to note that these determinant categories largely refer to the way an individual's "perceived" version of reality influences their behavior, rather than a theoretically objective state of reality which may not be in line with the individual's perceptions. This study sought to explore the behavior, "young unmarried women currently use modern contraceptive methods" among adolescent girls aged 15–19 in Kakamega County. For the purpose of this study, "modern contraceptive methods" was defined according to the Demographic and Health Surveys Program in the most recent DHS conducted in Kenya prior to the study, and it includes the following methods: female sterilization, male sterilization, the contraceptive pill, intrauterine contraceptive devices, injectables, implants, female condoms, male condoms, the lactational amenorrhea method, and the standard days method [22].

To identify those determinants that are the most powerful, the same types of questions are asked of "Doers" (those that do the behavior) and "Non-Doers" (those that do not do the behavior). The questions are open-ended, and the respondents provide qualitative responses; however, the structure of the questions results in large numbers of similar responses, and this enables the language of the responses to be standardized and analyzed quantitatively. During this quantitative analysis, percentage gaps between "Doers" and "Non-Doers" regarding individual responses are examined. Those that are the most significant show a gap of at least 15 percentage points between the percentages of Doers and Non-Doers who responded in a certain way. The Barrier Analysis approach recommends a sample size of 45 Doers and 45 Non-Doers in order to detect statistically significant odds ratios of 3.0 or higher at a confidence level of 90% [24]. For this Barrier Analysis, we

compiled 50 completed questionnaires from Doers and 46 completed questionnaires from Non-Doers.

*2.2. Study Setting and Participants*

The Barrier Analysis was undertaken in Navakholo, a sub-county of Kakamega County, Kenya, between November and December 2018.

Potential survey respondents were recruited using convenience sampling in the three sub-county communities that were purposively selected for the larger gender analysis. To identify young women for inclusion in the survey, the study capitalized on mobilization efforts for the larger study, including household visits by community health workers, as well as community-level recruitment opportunities, such as market days, health fairs, and project-related activities. The survey questionnaire included screening questions, so young women were screen as they were identified, and each young woman who agreed to participate was sorted into one of the two survey groups or excluded based on screening criteria. As a minimum of 90 participants were required, with a minimum of 45 participants in each survey group, enumerators were instructed to ensure a minimum of 30 participants from each of the three targeted communities, and to strive for a similar number of participants in each of the two groups. A total of 193 young women were identified and screened, and 96 were eligible to participate and sorted into one of the two survey groups.

Young women were eligible to participate in the study if they were aged 15–19 at the time of the study, were unmarried, were sexually active (defined as having had sex within the six months prior to the study), were not currently pregnant, had not given birth in the previous six months, and (a) provided informed and voluntary consent (if they were aged 18–19 or were emancipated minors) or (b) had parents/guardians who provided informed and voluntary consent if the young women were under 18 and not emancipated, after which the young women themselves provided informed and voluntary assent. Consent and assent were obtained in Swahili.

Eligible young women who met the above conditions were then categorized as Doers or Non-Doers of the study behavior using a single screening question. For respondents to be classified as a member of the "Doer" group, they had to indicate that they currently used a contraceptive method, which meant that the respondent personally was currently using a method, or that she and her partner used a condom every time they had sexual intercourse. Young women who responded that they were not currently using a contraceptive method were sorted into the "Non-Doer" group. Respondents received a different questionnaire based on their classification as Doer or Non-Doer, but were not told that they had been assigned into specific groups. Fifty young women completed the Doer questionnaire, and forty-six completed the Non-Doer questionnaire.

*2.3. Data Collection*

Data were collected by young female research assistants who received five days of training on research ethics, the study itself, and the questionnaire. They also received a reference manual. The Barrier Analysis questionnaire used the standardized question format for each determinant, and specific content related to the key behavior was developed with input from technical experts. The questionnaire (available in Supplementary Materials, file #) was developed in English and translated into Swahili, then back-translated into English to ensure accuracy. The Swahili version was pre-tested with eligible young women in communities on the outskirts of Kisumu town to mimic the conditions and respondents of the study itself. The Swahili and English questionnaires were revised as a result of the pre-test. The research assistants administered a tablet-based structured questionnaire to participants to collect Barrier Analysis data in a location of the respondent's choosing.

*2.4. Data Analysis*

The data were collected and managed using REDCap (Research Electronic Data Capture), a secure web-based electronic data collection platform with internal checks to support

data quality. The data were also reviewed by the study coordinator at the end of each day to minimize data collection and entry errors. REDCap data were downloaded into Excel. Open-ended responses were coded thematically, using both inductive and deductive coding. After this process, the coded responses were manually entered into the standardized Barrier Analysis tabulation spreadsheet (available in Supplementary Materials, file #), developed as part of the Designing for Behavior Change Framework, and checked by a second researcher to ensure that nothing was missed. The spreadsheet calculated the percentages of responses for Doers and Non-Doers, odds ratios, standard errors, confidence intervals, and *p*-values for each of the major determinant categories.

### 2.5. Ethical Considerations

The study received ethical approval from the Institutional Review Board at the Johns Hopkins University School of Public Health (IRB No 00008716) and the Amref Health Africa Ethics and Scientific Review Committee (AMREF-ESRC P455/2018). The study team strictly followed the study manual with a series of standard operating procedures. Written informed consent in Swahili was obtained from all adult and emancipated minor study participants, and parent/guardian written informed consent, with participant written informed assent, was obtained in Swahili from all non-emancipated minor participants.

## 3. Results

The study found noteworthy results across five major determinant categories: perceived self-efficacy, barriers, and enablers; perceived positive or negative consequences; perceived social norms; perceived susceptibility and severity; and universal motivators. In some cases, these results indicated statistically significant differences between Doers' and Non-Doers' responses, indicating that these responses represented key differences between the two groups and relevant barriers or enablers to contraceptive use. In other cases, the results were noteworthy because the researchers expected to see significant differences between Doers' and Non-Doers' responses, but those differences did not appear, indicating that expected behavioral determinants may not influence contraceptive behaviors among these populations.

### 3.1. Perceived Self-Efficacy, Barriers, and Enablers

Perceived self-efficacy is defined as a person's belief in their ability to do a behavior given the time, knowledge, skills, and resources available to them. Barriers and enablers are large determinant categories that encompass anything that makes it easier or more difficult to do a behavior. These barriers and enablers may or may not overlap with other determinant categories. Doers were almost 12 times more likely than Non-Doers to indicate that they did have the knowledge, skills, and resources to be able to use modern contraception, while Non-Doers were over four times more likely to indicate that they felt they did not have the knowledge, skills, and resources to be able to use modern contraception (Table 2). In addition, almost two thirds of Non-Doers (61%) indicated that they were unsure whether they would be able to use modern contraception given their current knowledge, skills, and resources; they were almost five times more likely than Doers to give this response. This could indicate that, rather than considering contraceptive use and finding it infeasible, or trying it and being unable to continue, Non-Doers may not have fully considered the practicalities of using modern contraception.

The possibility that Non-Doers may not have fully explored the process of using modern contraception is reinforced by the responses to questions about what makes it easier or harder to use contraception (Table 3). Almost half of Doers (44%) noted that contraceptives were easily accessible, with condoms and pills available for purchase at drug shops, and other methods available at health facilities. A further 26% of Doers noted that methods are easily available, and 16% mentioned that condoms and other methods are free or easily affordable. Contrary to this, 39% of Non-Doers stated that it would be easier to use contraceptives if they were more accessible or available, and 17% of Non-Doers

indicated that it would be easier to use contraceptives if they were more affordable or free. Seventeen percent of Non-Doers indicated that it would be easier to use modern contraception if they had additional information or guidance, while Doers did not mention access to information as either an enabler or a barrier. Finally, Non-Doers were two-and-a-half times more likely to be unable to provide any information on what would make it easier and only responded that they did not know.

**Table 2.** Contraceptive self-efficacy.

| | Contraceptive Self-Efficacy: Has Knowledge, Skills, and Resources to Use Contraception | | | | | | |
| --- | --- | --- | --- | --- | --- | --- | --- |
| **Response** | **Doers n (%)** | **Non-Doers n (%)** | **Diff** | **Odds Ratio** | **Confidence Interval** | **ERR** | ***p*-Value** |
| Yes | 38 (76%) | 7 (15%) | 61% | 17.64 | 6.28–49.60 | 11.70 | <0.001 |
| No | 3 (6%) | 11 (24%) | −18% | .20 | 0.05–0.78 | 0.22 | 0.013 |
| Don't know/maybe | 8 (16%) | 28 (61%) | −45% | 0.12 | 0.05–0.32 | 0.15 | <0.001 |

**Table 3.** Barriers and enablers to contraceptive use *.

| **Response** | **Doers n (%)** | **Non-Doers n (%)** | **Diff** | **Odds Ratio** | **Confidence Interval** | **ERR** | ***p*-Value** |
| --- | --- | --- | --- | --- | --- | --- | --- |
| Enabler: Contraceptives are easily accessible | 22 (44%) | 0 (0%) | 44% | N/A | N/A | N/A | N/A |
| Enabler: Contraceptives are easily available | 13 (26%) | 0 (0%) | 26% | N/A | N/A | N/A | N/A |
| Enabler: Contraceptives are free/easily affordable | 8 (16%) | 0 (0%) | 16% | N/A | N/A | N/A | N/A |
| Barrier: It would be easier to use contraceptives if they were more accessible | 0 (0%) | 12 (26%) | −26% | N/A | N/A | N/A | N/A |
| Barrier: It would be easier to use contraceptives if they were more available | 0 (0%) | 6 (13%) | −13% | N/A | N/A | N/A | N/A |
| Barrier: It would be easier to use contraceptives if they were free/more affordable | 0 (0%) | 8 (17%) | −17% | N/A | N/A | N/A | N/A |
| Barrier: It would be easier to use contraception if I had more information or guidance | 0 (0%) | 8 (17%) | −17% | N/A | N/A | N/A | N/A |
| Don't know what would make it easier to use contraception | 9 (18%) | 17 (37%) | −19% | 0.37 | 0.15–0.96 | 0.41 | 0.031 |

* For those variables which have zero observations for either the Doer or Non-Doer group, OR, CI, ERR, and *p*-value cannot be calculated, and only descriptive statistics are presented.

*3.2. Perceived Positive or Negative Consequences*

Perceived positive or negative consequences refer to an individual's belief in the advantages or disadvantages of doing the behavior, or the positive or negative outcomes that will occur as a result of doing the behavior. Almost all Doers (98%) noted that contraception prevents pregnancy, and many specified that it prevents early pregnancy (i.e., pregnancy at a young age), which is particularly relevant given the respondents' life stage. Seventy-eight percent of Non-Doers mentioned pregnancy prevention as a positive consequence of contraceptive use. While this does meet the threshold of a significant response (as there was a 20 percentage point difference between Doers and Non-Doers) it is

important to note that knowledge of pregnancy prevention is very high among both groups. The most common negative consequences mentioned were infertility and irregular or excessive bleeding. There was no significant difference between the two groups regarding concern about contraception causing infertility, and in the case of responses related to bleeding, more Doers than Non-Doers noted this to be an issue (42% of Doers compared with 17% of Non-Doers) (Table 4).

**Table 4.** Positive and negative consequences of using contraception.

| Response | Doers n (%) | Non-Doers n (%) | Diff | Odds Ratio | Confidence Interval | ERR | *p*-Value |
|---|---|---|---|---|---|---|---|
| Negative: Irregular bleeding | 21 (42%) | 8 (17%) | 25% | 3.44 | 1.33–8.87 | 2.92 | 0.008 |
| Negative: Causes infertility | 17 (34%) | 22 (48%) | −14% | 0.56 | 0.25–1.28 | 0.59 | 0.121 |
| Positive: Prevents pregnancy | 49 (98%) | 36 (78%) | 20% | 13.61 | 1.67–111.17 | 12.07 | 0.002 |

*3.3. Perceived Social Norms*

The determinant category of perceived social norms refers to an individual's belief in the social acceptability of doing a given behavior. Doers were almost three times more likely than Non-Doers to say that most people approve of their contraceptive use, while Non-Doers were twice as likely as Doers to say that most people would not approve (Table 5). Doers were four times more likely to indicate that their mothers approved of their contraceptive use, with 40% citing this approval. Doers were also four times more likely to indicate that their boyfriends approved of contraceptive use. When asked who disapproved, almost two-thirds (63%) of Non-Doers cited mothers, making them 2.5 times more likely than Doers to give this response. Over half of Non-Doers (57%) also indicated that no one would approve of them using contraception.

**Table 5.** Perceived social norms: who approves/would approve of my contraceptive use?

| Response | Doers n (%) | Non-Doers n (%) | Diff | Odds Ratio | Confidence Interval | ERR | *p*-Value |
|---|---|---|---|---|---|---|---|
| No one approves | 14 (28%) | 26 (57%) | −29% | 0.30 | 0.13–0.70 | 0.34 | 0.004 |
| Boyfriend approves | 10 (20%) | 2 (4%) | 16% | 5.50 | 1.14–26.63 | 3.98 | 0.020 |
| Mother approves | 20 (40%) | 5 (11%) | 29% | 5.47 | 1.84–16.21 | 4.17 | 0.001 |
| Most people I know disapprove | 22 (44%) | 29 (63%) | −19% | 0.46 | 0.20–1.04 | 0.50 | 0.048 |
| Most people I know approve | 16 (32%) | 6 (13%) | 19% | 3.14 | 1.10–8.91 | 2.68 | 0.024 |

*3.4. Perceived Susceptibility and Severity*

Perceived susceptibility refers to a person's perceived vulnerability to negative outcomes if they do not do the behavior, while perceived severity refers to the perceived seriousness of that negative outcome. Both Doers and Non-Doers indicated that it is very likely that they will become pregnant if they do not use modern contraception, with 88% of Doers and 80% of Non-Doers providing this response. All of the Doer respondents and 93% of the Non-Doer respondents also indicated that it would be a very serious problem for them if they became pregnant (Table 6).

**Table 6.** Perceived susceptibility and severity.

| Response | Doers n (%) | Non-Doers n (%) | Diff | Odds Ratio | Confidence Interval | ERR | *p*-Value |
|---|---|---|---|---|---|---|---|
| It is likely I will get pregnant if I do not use contraception | 44 (88%) | 37 (80%) | 8% | 1.78 | 0.58–5.48 | 1.67 | 0.230 |
| It would be a very bad thing if I got pregnant | 50 (100%) | 43 (93%) | 7% | | | | 0.106 |

*3.5. Universal Motivators*

Universal motivators refer to the set of motivators that influence most people, regardless of context. When asked, "What do you desire most in life?", many respondents indicated a desire to finish school, get a job, and generally have a good life. Only 8% of Doers and 4% of Non-Doers specifically mentioned the life goal of wanting to have a family (though this may be due to the respondents providing answers that they expected the enumerators wanted to hear, or wanting to seem sophisticated to the young women who were conducting the research). Over three quarters of Non-Doers (76%) mentioned a goal of having a specific job (such as being a doctor, teacher, journalist, artist, or police officer), compared with roughly a third (34%) of Doers (Table 7).

**Table 7.** Universal motivators.

| Response | Doers n (%) | Non-Doers n (%) | Diff | Odds Ratio | Confidence Interval | ERR | *p*-Value |
|---|---|---|---|---|---|---|---|
| I want to have a family | 4 (8%) | 2 (4%) | 4% | 1.91 | 0.33–10.98 | 1.76 | 0.379 |
| I want to be a (specific job) | 17 (34%) | 35 (76%) | −42% | 0.16 | 0.07–0.40 | 0.20 | <0.001 |

**4. Discussion**

Our study used the Barrier Analysis methodology to identify differences between sexually active young unmarried women who were either contraceptive users or non-users (identified as Doers and Non-Doers in this article). Our analysis revealed that Doers were more likely than Non-Doers to perceive themselves as having the knowledge, skills, and resources needed to use contraception. Non-Doers were more likely to express uncertainties about whether or not they had the necessary knowledge, skills, and resources. The availability of information about contraceptives is a critical driver for increased uptake, as was evidenced by an analysis of demographic and health surveys conducted in sub-Saharan Africa which revealed that knowledge of modern contraceptives had associations with the utilization of modern contraceptives among adolescent girls and young women [25]. However, though information is necessary, it is not sufficient when other barriers to uptake exist.

Doers were also more likely than Non-Doers to report ease of access to affordable contraceptives. While it is possible that Non-Doers may face additional resource, geographic, or mobility constraints, and that these might account for the above differences, it could also be that their *perception* of the accessibility, availability, and cost constraints is different, rather than their reality. This may be driven by a lack of practical information on where to access contraceptive services or their cost, leading to a perceived inability to afford them [26]. It is well documented that in Kenya, the majority of adolescents prefer accessing contraceptive services through private facilities, such as pharmacists and drug shops [21]. These channels often charge for contraceptive services, which creates barriers, as most adolescents may not have the resources to purchase contraceptive commodities and services. In addition, adolescents face other challenges when it comes to contraceptive service provision due to provider bias and misconceptions [27], and these further limit their access to services.

The study respondents indicated concerns about side effects (both real, such as bleeding changes, and misconceptions, such as risk of infertility) as negative consequences of contraceptive use. However, it is important to note that both Doers and Non-Doers cited these same issues, and yet Doers continue to use modern contraception. This suggests that these issues are not key reasons why a young woman in Kakamega would decide against using modern contraception. Studies have revealed that, at the global level, contraceptive side effects, either experienced or perceived, are a major deterrent to the uptake or continued use of contraceptives [28,29]. However, our study finds a different result in the context of Kakamega, in which users are more likely than their non-using peers to identify side effects as a negative consequence to using contraceptives, but this does not deter them. These findings suggest that for this population, other factors are more important in determining contraceptive use, such as those noted in the study. Additional research may be beneficial and lead to a better understanding of why these negative consequences do not have the same influence over young women's behaviors in Kakamega as has been reported in other contexts. Such findings could be relevant beyond Kakamega County, or Kenya in general, and beyond adolescents to women of reproductive age.

Concerning perceived social norms, Doers were more likely to report societal approval of contraceptive use than Non-Doers. The question of family and peer approval represents the greatest difference in responses between young women who use contraception and those who do not. It may be that there is something different about the mothers and boyfriends of Non-Doers that make them unwilling to approve of contraceptive use in young women. For example, they may have less exposure to information about contraception, or hold more conservative views about contraceptive use. It is also possible that the mothers and boyfriends of Non-Doers would actually approve of their daughters/girlfriends using contraception, but the Non-Doer young women believe that they would disapprove, and so have never brought up the issue.

Both Doers and Non-Doers reported high perceived susceptibility to becoming pregnant if they don't use contraceptives. Similarly, both groups agree that becoming pregnant would be a significant problem. These findings are corroborated by a study conducted in Kenya which concluded that adolescents fear pregnancy acutely due to stigma that could result in social retribution [30]. Despite this, Non-Doers are still not motivated by fear of pregnancy to use contraception, which suggests that messaging that indicates that getting pregnant will negatively impact their lives in different ways would not be effective. On the other hand, the fact that a large percentage of Non-Doers had specific goals they were working towards could provide an opportunity for an intervention that linked those goals to contraceptive use.

Our findings should be interpreted with the following limitations in mind. The Barrier Analysis approach does not incorporate inferential statistics to identify the determinants for contraceptive use, a factor which could further explain the differences between women who use contraceptives and those who do not. For instance, other studies have shown that age, education, and working status are significantly associated with contraceptive use among adolescent girls [31], and that educated mothers are more open to discussing family planning with their children [32]. There may also be differences between those young women who opted into the study versus those who did not, or between young women who self-consented versus those who required parent/guardian consent. It is also worth noting that some CIs may be wide because of the wide variations in sample size within the observations per category, as well as the overall small sample size. Finally, as with most qualitative studies, the results may not be generalizable beyond the specific study population. Despite these limitations, our findings highlight important differences between contraceptive users and non-users among young, sexually active, unmarried women.

## 5. Conclusions

Our findings indicate differences among young female contraceptive users and non-users in Kakamega County across three major determinant categories: perceived self-

efficacy, barriers, and enablers; perceived social norms; and universal motivators. Our findings also indicate a lack of differences (or unexpected differences) for two other key determinant areas: perceived positive and negative consequences; and perceived susceptibility and severity of consequences of becoming pregnant. These findings can be used to create contextually relevant and effective programming to increase access to and use of contraceptives among young women. Based on these results, the authors recommend the following programmatic strategies for strengthening contraceptive use among unmarried young women in Kakamega, and in similar contexts:

- Ensure that all community outreach and communication activities include practical information about where contraception is available and its cost, using means that are trusted and accessible to unmarried young women.
- Engage the mothers and boyfriends of contraceptive users to act as contraceptive champions, and to provide testimonials and engage with their peers to share personal stories about how modern contraceptive use has positively impacted their lives and the lives of their daughters/girlfriends; and/or engage young women who use contraception and have the support of their mothers and/or boyfriends to share their stories with non-users to illustrate that family support is feasible and likely.
- Use human-centered designs to co-create positive, motivational messages with users and non-users about how contraceptive use can help young woman achieve their life goals.

**Supplementary Materials:** The study questionnaire and dataset can be downloaded at: https://www.mdpi.com/article/10.3390/adolescents3030026/s1.

**Author Contributions:** Conceptualization, E.A.-P. and C.A.; data curation, E.A.-P.; formal analysis, E.A.-P. and C.A.; writing—original draft preparation, E.A.-P., C.A., S.O. and L.M.; writing—review and editing, E.A.-P., C.A., S.O. and L.M. All authors have read and agreed to the published version of the manuscript.

**Funding:** The study was funded by USAID Kenya and East Africa under the Afya Halisi project, award number AID-615-A-17-00004. The funding institution did not play a role in the study design or implementation, in the writing of the manuscript, or in the decision to submit the article for publication. The views and opinions expressed in this paper are those of the authors and are not necessarily the views and opinions of USAID.

**Institutional Review Board Statement:** This study was conducted in accordance with the Declaration of Helsinki and approved by the Institutional Review Board at the Johns Hopkins University School of Public Health (IRB No. 00008716; 25 July 2019), the Amref Health Africa Ethics and Scientific Review Committee (AMREF-ESRC P455/2018; 28 May 2018), and the Kakamega County Government.

**Informed Consent Statement:** Written informed consent was obtained from all adult and emancipated minor subjects who participated in the study. Caregivers of adolescents provided written informed consent for non-emancipated adolescent subjects, after which the adolescents provided written assent.

**Data Availability Statement:** The data presented in this study are available as part of the Supplementary Materials and can be accessed at the following link: www.mdpi.com/xxx/s1.

**Acknowledgments:** The authors would like to thank the project donor, USAID, for providing funds to support this study. We are grateful to the Kakamega County and Navakholo Sub-County Health Management Committees and the Gender Directorate for their insights into the design of the study. We are grateful to the study coordinators and research assistants for their commitment and excellent work during data collection, to the Afya Halisi Monitoring and Evaluation team for putting standards in place to ensure data quality and completeness, and to Myra Betron and Meghan Greeley, Jhpiego Baltimore, and Lilian Mutea, USAID Kenya, for their review of the manuscript. Finally, we express our gratitude to all the study respondents from Kakamega who participated in this study and to the parents and caregivers who allowed us to engage their children on the subject matter.

**Conflicts of Interest:** The authors declare no conflict of interest. The funders had no role in the design of the study; in the collection, analyses, or interpretation of data; in the writing of the manuscript; or in the decision to publish the results.

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
