# Peer review of "Determinants of Contraceptive Use among Unmarried Young Women in Kakamega County, Kenya"

_adolescents, doi:10.3390/adolescents3030026_

Round 1
Reviewer 1 Report
Comments and Suggestions for Authors
Manuscript ID: Adolescents – 2311364
This is an interesting paper on an especially important public health topic on adolescent sexual and reproductive health in Kenya. Below comments are aimed to improve the paper.
Abstract
The authors should present some of the quantitative data (e.g. numbers, percentages, odds ratios, and p-values) to strengthen the findings.
Introduction
Reference the following sentences:
· Adolescents age 10 to 19 constitute 16% of the world’s population.
· Teenage pregnancy and harmful socio-cultural practices constitute some of the determinants of early marriages.
· Kenya’s National Adolescent Sexual and Reproductive Health policy outlines adolescents’ sexual and reproductive rights and a framework to govern service provision.
Methods
Under the study procedure and measures section, the authors stated that they used “standardized and validated questions” presented in Table 1. However, under data collection, they stated that the “tool was created with inputs from technical experts and underwent review and revision through in-country”. These two contradict and I suggest that the authors should clarify. In addition, they should give more details about the pre-testing of the tool. Also, it seems as if the translated questionnaire into Swahili was never pre-tested/piloted. The authors must also indicate the language used for the informed and voluntary consent, which I presume was in English.
Results
Whilst the results are written in simple and clear sentences, the authors must include the odds ratios, standard errors, confidence intervals and p values where applicable so that readers can identify them from the tables. For example, I struggled to identify “four times” from Table 2 in the following sentence, “…while Non-Doers were over four times more likely than Doers to indicate that they felt they did not have the knowledge, skills, and resources to be able to use modern contraception (Table 2).”
The authors must rephrase some sentences as other parts are not in the questionnaire/supplementary material. For example, I could not find the underlined parts in the following statements.
· Almost half of Doers (44%) noted that contraceptives were easily accessible, with condoms and pills available for purchase at drug shops, and other methods available at health facilities.
· A further 26% of Doers noted that methods are easily available and 16% mentioned that condoms and other methods are free or easily affordable.
Again, it is not clear why the authors presented the odds ratios and p values in Table 3, whilst there is mention of them in the description. The authors should also consider presenting the significant differences in Table 4 and other tables.
Wishing you all the best.
Author Response
Please see below for the responses to your comments. The revised manuscript with revisions in tracked changes is attached.
Abstract: The authors should present some of the quantitative data (e.g. numbers, percentages, odds ratios, and p-values) to strengthen the findings. |
Quantitative data has been included in the abstract. |
Introduction: Please reference the following sentences: · Adolescents age 10 to 19 constitute 16% of the world’s population. · Teenage pregnancy and harmful socio-cultural practices constitute some of the determinants of early marriages. · Kenya’s National Adolescent Sexual and Reproductive Health policy outlines adolescents’ sexual and reproductive rights and a framework to govern service provision. |
These references have been included. |
Methods: Under the study procedure and measures section, the authors stated that they used “standardized and validated questions” presented in Table 1. However, under data collection, they stated that the “tool was created with inputs from technical experts and underwent review and revision through in-country”. These two contradict and I suggest that the authors should clarify. |
This has been clarified to note that the structure of each question is standardized, and includes room for specificity based on the key behavior being examined. It is this behavior-related content that was created with inputs from technical experts and reviewed and revised in-country. Please see lines 112-118 and 184-185. |
Methods: In addition, they should give more details about the pre-testing of the tool. Also, it seems as if the translated questionnaire into Swahili was never pre-tested/piloted. The authors must also indicate the language used for the informed and voluntary consent, which I presume was in English. |
Additional details have been added regarding pre-testing of the tool. It has been clarified that the tool was pre-tested in Swahili, and consent/assent as sought in Swahili. Please see lines 170 and 186-191. |
Results: Whilst the results are written in simple and clear sentences, the authors must include the odds ratios, standard errors, confidence intervals and p values where applicable so that readers can identify them from the tables. For example, I struggled to identify “four times” from Table 2 in the following sentence, “…while Non-Doers were over four times more likely than Doers to indicate that they felt they did not have the knowledge, skills, and resources to be able to use modern contraception (Table 2).” |
These statements of increased or decreased likelihood are based on the estimated relative risk (ERR). Thus, an ERR of 11.70 translates into the description of the first part of the sentence in question, “Doers were almost 12 times more likely than Non-Doers to indicate that they did have current knowledge, skills, and resources…” The second part of the sentence (mentioned in the comment) uses the ERR of 0.22. This ERR is discussed from the Non-Doer point of view, so rather than noting that Doers are over four times less likely than Non-Doers to respond in this way, the paper notes that Non-Doers are over four times more likely than Doers to respond this way. |
Results: The authors must rephrase some sentences as other parts are not in the questionnaire/supplementary material. For example, I could not find the underlined parts in the following statements. · Almost half of Doers (44%) noted that contraceptives were easily accessible, with condoms and pills available for purchase at drug shops, and other methods available at health facilities. · A further 26% of Doers noted that methods are easily available and 16% mentioned that condoms and other methods are free or easily affordable. |
This comment was received without any underlining in the statements. We have asked the journal to reach back out to the reviewer to clarify the question. |
Results: Again, it is not clear why the authors presented the odds ratios and p values in Table 3, whilst there is mention of them in the description. The authors should also consider presenting the significant differences in Table 4 and other tables. |
Table 3 has been revised to note that, for those responses in which one category (Doer or Non-Doer) includes zero observations, only descriptive statistics have been presented.
|
Reviewer 2 Report
Comments and Suggestions for Authors
ABSTRACT
Authors should define modern contraception
INTRODUCTION
Very thorough introduction!
METHODS
It is not clear what is meant by "modern contraceptive methods". Does this mean women who did not use modern methods were excluded?
Very interesting use of Barrier analysis
RESULTS
The CI for the Contraceptive Self Efficacy Odds Ratio and Prevents pregnancy were quite wide. Measures to account for this should be detailed.
Consistent number of decimal places should be used throughout
Author Response
Please see below for responses to your comments.
ABSTRACT: Authors should define modern contraception |
Given the word limit, the definition of modern contraceptive methods has been included in the methods section. The study uses the Demographic and Health Surveys definition of modern methods of contraception that was used in the most recent DHS conducted in Kenya prior to the Barrier Analysis study. Please see lines 126-132. |
INTRODUCTION: Very thorough introduction!
|
Thank you. |
METHODS: It is not clear what is meant by "modern contraceptive methods". Does this mean women who did not use modern methods were excluded? |
“Modern contraceptive methods” have been defined in the Methods section using the definition outlined by the Kenya Demographic and Health Surveys Program. Please see lines 126-132.
As noted in the Methods section, if a young woman met all of the screening criteria, and was using a method, she was classified as a “Doer,” while if she met all of the screening criteria and was not using a method, she was classified as a “Non-Doer.” |
Methods: Very interesting use of Barrier analysis |
Thank you. |
Results: The CI for the Contraceptive Self Efficacy Odds Ratio and Prevents pregnancy were quite wide. Measures to account for this should be detailed. |
It is expected that some CI may be wide because of the wide variation of sample size within the observations per category, as well as the overall small sample size. We have added an acknowledgement of this issue as part of the limitations in the Discussion section. Please see lines 446-448. |
Results: Consistent number of decimal places should be used throughout |
This has been adjusted so that odds ratios, confidence intervals, and ERRs have been listed to two decimal places. P values continue to be listed to three decimal places. |
Reviewer 3 Report
Comments and Suggestions for Authors
This is a really interesting paper, and I am glad I have a chance to read and review it. I have just a few questions, which I think can be easily addressed.
In the background section, I am not sure what this statement is trying to say: “..which had the goal of increasing the use of quality reproductive, maternal, newborn, child and adolescent health..”. I think there is just a word missing (“services”, maybe?).
In the Methods section, it is not noted how study participants were identified? How were they sampled?
In the Methods section, I would like to see more details on the analysis. For example, how many people coded the qualitative data in order to develop the codes? Were there discrepancies? If so, how were those resolved? Was there an audit trail?
I am not familiar with the Barrier Analysis methodology, but I’m surprised that these quantified qualitative data are not able to build an inferential model. While the authors note that this in an inappropriate way to handle these sorts of data, I would like to see a deeper explanation as to why this is. As a reader hearing about this method for the first time, I was not sure why a model couldn’t be built.
Author Response
Please see below for responses to your comments.
This is a really interesting paper, and I am glad I have a chance to read and review it. I have just a few questions, which I think can be easily addressed. In the background section, I am not sure what this statement is trying to say: “..which had the goal of increasing the use of quality reproductive, maternal, newborn, child and adolescent health..”. I think there is just a word missing (“services”, maybe?). |
This has been rephrased, including the addition of the missing word, “services.” Please see lines 92-96.
|
In the Methods section, it is not noted how study participants were identified? How were they sampled? |
Additional information has been added to the “Study setting and participants” section regarding the identification and sampling process. Please see lines 149-162. |
In the Methods section, I would like to see more details on the analysis. For example, how many people coded the qualitative data in order to develop the codes? Were there discrepancies? If so, how were those resolved? Was there an audit trail? |
One person coded the qualitative responses, and the resulting coded data was sense-checked by a second person to ensure that nothing was missed. This has been added into the Methods section. Please see lines 218-219. |
I am not familiar with the Barrier Analysis methodology, but I’m surprised that these quantified qualitative data are not able to build an inferential model. While the authors note that this in an inappropriate way to handle these sorts of data, I would like to see a deeper explanation as to why this is. As a reader hearing about this method for the first time, I was not sure why a model couldn’t be built. |
The Barrier Analysis methodology uses a predetermined analysis approach that does not include inferential statistics. In addition, it should be clarified that our use of the term “determinants” refers to the social science usage rather than the research usage which infers the use of regression analysis. |